# A Calibration/Disaggregation Coupling Scheme for Retrieving Soil Moisture at High Spatio-Temporal Resolution: Synergy between SMAP Passive Microwave, MODIS/Landsat Optical/Thermal and Sentinel-1 Radar Data

**DOI:** 10.3390/s21217406

**Published:** 2021-11-08

**Authors:** Nitu Ojha, Olivier Merlin, Abdelhakim Amazirh, Nadia Ouaadi, Vincent Rivalland, Lionel Jarlan, Salah Er-Raki, Maria Jose Escorihuela

**Affiliations:** 1CESBIO, Université de Toulouse, CNES/CNRS/INRA, IRD/UPS, 31400 Toulouse, France; ojhan@cesbio.cnes.fr (N.O.); nadia.ouaadi@gmail.com (N.O.); vincent.rivalland@cesbio.cnes.fr (V.R.); lionel.jarlan@ird.fr (L.J.); 2Center for Remote Sensing Applications (CRSA), Mohammed VI Polytechnic University (UM6P), Ben Guerir 43150, Morocco; abdelhakim.amazirh@gmail.com; 3LMFE, Department of Physics, Faculty of Sciences Semlalia, Cadi Ayyad University, Marrakech 40000, Morocco; 4ProcEDE, Département de Physique Appliquée, Faculté des Sciences et Techniques, Université Cadi Ayyad, Marrakech 40000, Morocco; s.erraki@uca.ma; 5isardSAT S.L., Parc Tecnologic Barcelona Activa, 08042 Barcelona, Spain; mj.escorihuela@isardsat.cat

**Keywords:** disaggregation, soil moisture, synergy, Sentinel-1, DISPATCH, SMAP, Landsat

## Abstract

Soil moisture (SM) data are required at high spatio-temporal resolution—typically the crop field scale every 3–6 days—for agricultural and hydrological purposes. To provide such high-resolution SM data, many remote sensing methods have been developed from passive microwave, active microwave and thermal data. Despite the pros and cons of each technique in terms of spatio-temporal resolution and their sensitivity to perturbing factors such as vegetation cover, soil roughness and meteorological conditions, there is currently no synergistic approach that takes advantage of all relevant (passive, active microwave and thermal) remote sensing data. In this context, the objective of the paper is to develop a new algorithm that combines SMAP L-band passive microwave, MODIS/Landsat optical/thermal and Sentinel-1 C-band radar data to provide SM data at the field scale at the observation frequency of Sentinel-1. In practice, it is a three-step procedure in which: (1) the 36 km resolution SMAP SM data are disaggregated at 100 m resolution using MODIS/Landsat optical/thermal data on clear sky days, (2) the 100 m resolution disaggregated SM data set is used to calibrate a radar-based SM retrieval model and (3) the so-calibrated radar model is run at field scale on each Sentinel-1 overpass. The calibration approach also uses a vegetation descriptor as ancillary data that is derived either from optical (Sentinel-2) or radar (Sentinel-1) data. Two radar models (an empirical linear regression model and a non-linear semi-empirical formulation derived from the water cloud model) are tested using three vegetation descriptors (NDVI, polarization ratio (PR) and radar coherence (CO)) separately. Both models are applied over three experimental irrigated and rainfed wheat crop sites in central Morocco. The field-scale temporal correlation between predicted and in situ SM is in the range of 0.66–0.81 depending on the retrieval configuration. Based on this data set, the linear radar model using PR as a vegetation descriptor offers a relatively good compromise between precision and robustness all throughout the agricultural season with only three parameters to set. The proposed synergistical approach combining multi-resolution/multi-sensor SM-relevant data offers the advantage of not requiring in situ measurements for calibration.

## 1. Introduction

Soil moisture (SM) controls the energy exchange between the land surface and the atmosphere, as well as the terrestrial hydrological cycle and ecological environments [1]. Hence, SM information at a fine space-time scale is beneficial for agricultural [2] and other hydrological applications [3]. In situ measurements can provide SM estimates at a field scale, but they cannot capture all the spatial variability. The spatial variability of SM especially occurs due to vegetation, soil roughness, soil texture, terrain, and atmospheric and anthropogenic (e.g., irrigation) effects, which significantly impact SM [4]. One major motive for monitoring SM at a fine scale is the management of water resources over irrigated areas, i.e., the optimization of irrigation in terms of scheduling and dispatching [5,6]. Collecting time and cost are also the main constraints that make in situ SM measurements impractical for global SM monitoring. Instead, remote sensing offers a good compromise in the global tracking of SM over an enormous range of spatial and temporal scales.

Nowadays, L-band microwave radiometers are routinely used to provide SM data on a global basis. Based on L-band radiometry, two satellites are currently in operation: (1) Soil Moisture and Ocean Salinity (SMOS), launched by ESA in November 2010 [7], and (2) Soil Moisture Active Passive (SMAP), launched by NASA in January 2015 [8]. Both satellites provide SM retrievals at about 40 km resolution with a sensing depth of 3 to 5 cm and a global revisit cycle of 3 days. The L-band radiometry is one of the optimal technologies widely accepted for SM estimation [9]. SMOS/SMAP SM products have been extensively validated and found suitable for climatology and large-scale hydrology purposes. Nonetheless, their typical spatial resolution of 40 km is too coarse for most hydro-agricultural applications [10,11,12].

Other remote sensing techniques such as radar or thermal imagery can provide SM information at a much higher spatial resolution than L-band radiometers. For instance, Landsat-8 thermal and Sentinel-1 C-band radar sensors achieve a spatial resolution of 100 and 20 m, respectively. Based on the assumption that the passive microwave-derived SM is accurate at low resolution and that relative SM information can be obtained at higher spatial resolution from radar/thermal sensors, various disaggregation approaches have been proposed [13,14]. On the one hand, the radar-based disaggregation technique combines the low-resolution L-band brightness temperature with fine resolution radar observations. In this vein, the SMAP satellite was originally dedicated to combining an L-band radiometer and an L-band radar to provide SM at 3 km resolution. However, due to the SMAP radar failure, currently, the SMAP mission provides SM at 9 km resolution on a global basis [15] by interpolating the brightness temperature of the L-band radiometer using the Backus-Gilbert method [16]. Another approach used by the SMAP mission is to combine the SMAP radiometer with the Sentinel-1 radar to provide SM at 9 and 3 km [17]. However, this approach is limited by the constraint of the need for quasi-simultaneous overlapping areas of SMAP and Sentinel-1 data.

On the other hand, the optical/thermal-based downscaling method generally relies on the relationship between passive microwave-derived SM and the evaporative fraction derived from the vegetation index (NDVI)-land surface temperature (LST) feature space [13,18,19,20]. The main advantage of optical-based over radar-based downscaling approaches is that the LST is less affected by the soil roughness [21] and vegetation structure [22] than the radar backscatter coefficient [23,24,25]. However, optical-based approaches are limited by (1) the low repeat cycle of currently available high-spatial-resolution thermal sensors, (2) the gaps in data coverage due to the presence of clouds, and (3) the underlying requirement of a large atmospheric evaporative demand at the time of thermal sensor overpass.

Radar data are primarily influenced by surface roughness [26], vegetation cover [27] and topography [28], making the direct estimation of SM from backscattering coefficients more difficult than from the SMOS/SMAP L-band brightness temperature. However, active microwave sensors are sensitive to SM [29,30] and C-band synthetic aperture radars (SAR) have the capability to provide SM data at high spatial resolution with a global time coverage of 6–12 days with Sentinel-1, regardless of cloud coverage [31,32,33]. Various soil backscattering models [34], empirical and semi-empirical models [24,35,36] were developed to simulate the radar backscatter in the forward mode and to estimate SM from radar data in the retrieval mode. Because of water content in both the soil and vegetation components, estimating SM from the radar backscattering coefficient is a very complex task. The Water Cloud Model (WCM) [37] was developed to simulate the backscattering of the canopy. WCM is a simple mathematical model, and due to its simplicity, it is widely used in the forward or inverse mode [31,38,39,40,41]. The vegetation parameter in the WCM can be quantified using various vegetation descriptors such as the normalized difference vegetation index (NDVI), leaf area index (LAI), polarization ratio (PR), and the interferometric coherence (CO) [42,43,44].

Despite the pros and cons of the above SM retrieval techniques in terms of spatio-temporal resolution and their sensitivity to perturbing factors such as vegetation cover, soil roughness and meteorological conditions, there is currently no synergistic approach that takes advantage of all relevant remotely sensed data. To overcome the above-discussed limitations, this study aims to develop a new algorithm to provide SM at high spatio-temporal resolution without the need for in situ SM measurements for calibration of radar data. To do this, the new algorithm combines multi-resolution passive microwave, active microwave and optical/thermal data and an innovative calibration approach. As a first step, the DISPATCH algorithm is implemented at 100 m resolution to disaggregate the 36 km resolution SMAP SM at 100 m resolution on clear sky days [45]. DISPATCH is one of the reference disaggregation methods based on optical/thermal data [13,14,46,47]. As a second step, the 100 m resolution disaggregated SMAP SM data set is used to calibrate—on clear sky days with quasi-simultaneous overpasses of SMAP, Landsat, and Sentinel-1—two different C-band radar models: an empirical linear regression and a semi-empirical non-linear model based on the WCM formulation. In both (linear and non-linear) cases, three configurations are tested based on different vegetation descriptors in the model: the NDVI, PR, and CO separately. The different vegetation descriptors are used to evaluate the performance of each vegetation descriptor within the calibration/validation approach of the radar-based SM retrieval scheme. As a third and last step, both calibrated radar models are run in the inverse mode on all Sentinel-1 overpass dates to estimate the fine-scale SM at the temporal frequency of Sentinel-1.

Note that most past studies that have investigated soil moisture-related remote sensing tools such as SMAP, MODIS, Landsat and Sentinel-1, have compared the relative performance of individual techniques (e.g., [14,48,49]). Nevertheless, none of them have combined all the above sensors within a unique and spatially consistent method. To the knowledge of the authors, this study is the first to make synergistic use of SMAP, MODIS, Landsat and Sentinel-1 data to produce a single soil moisture data set at high spatio-temporal resolution. At the same time, recent research progresses present good direction for downscaling SMAP-like data using ancillary optical/thermal data [50,51,52]. However, such disaggregation approaches are still generally implemented at the 1 km resolution using MODIS or Sentinel-3 data [53,54,55,56]. Our study fundamentally differs from those previous approaches in that DISPATCH is implemented at 100 m resolution by using Landsat data, so that the spatial variability of SM is represented at a much finer scale, which is now consistent with the typical size of crop fields.

The proposed original disaggregation/calibration method for SM retrieval is tested over irrigated and rainfed wheat crop sites in the Haouz plain, central Morocco. In particular, the SM predicted by both models in different configurations are compared over the study area to analyze which retrieval approach performs better during the different stages of the agricultural season.

## 2. Materials and Methods

### 2.1. Study Area and In Situ Data

The study area comprises three experimental sites, namely the R3 irrigated zone and two (Sidi Rahal and Chichaoua) experimental crop fields, all located in the Haouz plain within a 100 km distance of Marrakesh city (see Figure 1). The Haouz plain has a semi-arid Mediterranean climate with an average annual precipitation of 250 mm [57,58] and an average evaporative demand of about 1600 mm/year [59]. The soil texture is clayey for the R3 irrigated zone, sandy for the Sidi Rahal site, and loamy-clayey for the Chichaoua site. The land of three experimental sites (R3 irrigated zone, Sidi Rahal and Chichaoua) are covered by agricultural crops, primarily cultivated with winter wheat.

The R3 irrigated study zone contains 22 irrigated (flood-irrigated) wheat parcels, with 3–4 ha each. The strategy for soil moisture sampling over R3 is explained in Amazirh et al. [21] and Ojha et al. [45]. In each of the 22 crop fields, 10 separate theta probe measurements were undertaken with 5 on a side of the field and 5 on the other side, by making sure that all measurements were taken sufficiently far (>5 m) from the field border for spatial representativeness issues. The field-scale in situ soil moisture data used in this paper were obtained by (1) calibrating the theta probe readings using gravimetric measurements and (2) averaging the 10 measurements for each field. Regarding the soil texture analysis, the samples were packed in plastic bags and properly marked for identification and analysis. A total of 20 g of the mixed soil was sampled for analyzing grain size distribution. These soil samples were air-dried and sieved into two fractions: (0.05–2 mm) to calculate the percentage of sand content. The smaller fraction passed through a (0.05 mm) sieve was recovered and collected in a vial and then analyzed using pipette [60] and/or granulometry laser methods to measure coarse loam (20–50 μm), fine loam (2–20 μm) and clay content (<2 μm). In this study, five sampling days (day of year 14, 30, 38, 62, 78) in 2016 are used concurrently with Sentinel-1 overpasses.

Sidi Rahal and Chichaoua sites cover an area of 1 and 1.5 ha separately. In situ SM data were collected in both the sites every 30 min using time domain reflectometry (TDR) sensors. Sidi Rahal is a rainfed wheat site, and Chichaoua a drip-irrigated wheat site. At the Chichaoua site, two TDR sensors were mounted with one between and one under the drippers. The average TDR measured values were calibrated using the gravimetric method. More detailed information about the in situ data collection is presented in Rafi et al. [58], Amazirh et al. [21], Ouaadi et al. [61], Ait Hssaine et al. [62]. The SM data collected during the time period of 2017–2018 were used in this paper. Note that the irrigated crop in Chichaoua underwent controlled water stress during the 2018 agricultural season [58].

Note that data from Chichaoua and Sidi Rahal sites were used for calibration and validation in this paper. However, R3 irrigated zone datasets were used only for validation.

### 2.2. Remote Sensing Data

#### 2.2.1. SMAP

The SMAP satellite was launched on 31 January 2015 by NASA [8]. SMAP is the first L-band satellite dedicated to provide SM at a resolution ranging from 3 to 36 km with a 2–3-day revisit cycle by incorporating both radar and radiometer. However, due to the failure of the SMAP radar, the SM generated by the SMAP processing chains is now defined at 36 and 9 km (by resampling technique) resolution. SMAP is a near-polar sun-synchronous orbit at an altitude of 658 km, with a descending/ascending overpass at 6:00 a.m./p.m. local time. To manage the SMAP radar failure, the SMAP mission has recently provided a product that combines SMAP and C-band Sentinel-1 radar data to provide an SM product at 1 km resolution. In this paper, the 36 km resolution SMAP level-3 version 005 product available on an EASE grid 2 at https://nsidc.org/data/SPL3SMP/versions/5 for a time period of 2016 to 2018 is used as input to the DISPATCH disaggregation method to provide SM data at 100 m resolution over the study areas.

#### 2.2.2. MODIS

MODIS version 6 MOD11A1 and MOD13A2 products on ascending mode Terra overpass (10:30 a.m.) and MODIS version 6 MYD11A1 product on descending mode Aqua overpass (1 p.m.) were used in this paper. LST and enhanced vegetation index (EVI) data extracted from MOD11A1/MYD11A1 and MOD13A2 products, respectively, were used as input to the DISPATCH disaggregation method. In practice, the C4DIS processor is applied to SMAP level-3 data to disaggregate SM data at 1 km resolution [54,63].

#### 2.2.3. Landsat

In this paper, optical/thermal data from Landsat-7 and Landsat-8, which have an offset of 8 days, were used as an input to the DISPATCH disaggregation method. The DISPATCH algorithm is applied to 1 km resolution disaggregated SMAP SM data using 100 m resolution Landsat LST/NDVI data to produce a disaggregated SM data set at 100 m resolution [45]. Landsat images are downloaded from the USGS website, which provides thermal radiance and surface reflectance data on a 30 m resolution sampling grid globally. Surface reflectance is aggregated at 100 m resolution and used to derive NDVI. The Landsat thermal radiance is used for the LST calculation using a single band algorithm from band-6 and band-10 of Landsat-7 and Landsat-8, respectively. The detailed description of the Landsat processing [64] is described in Ojha et al. [45].

#### 2.2.4. Sentinel-1

Sentinel-1 is a C-band SAR mission that consists of two satellites, Sentinel-1 A (S1-A) launched in April 2014 and Sentinel-2 (S2-B) launched in April 2016. Sentinel-1 is a sun-synchronous near-polar orbit satellite with a revisit cycle of 12 days. In this paper, the radar-based SM retrieval models use VV polarization. As stated in Amazirh et al. [21] and Ouaadi et al. [31], VV polarization is a better indicator of SM than VH for the selected study area. Note that VH polarization is still used herein to estimate the PR defined as the ratio of backscattering coefficients in VH and VV polarizations.

In this paper, the Ground Range Detected (GRD) and Single Look Complex (SLC) products were downloaded from the Copernicus Sentinel hub. GRD products were used to compute the backscattering coefficient, whereas SLC products were used to calculate interferometric coherence. Sentinel-1 level-1 GRD products were processed in four steps: (1) removing thermal noise by removing the additive effect, (2) radiometric calibration to compute the backscattering coefficient by using sensor calibration parameters, (3) terrain correction by correcting the backscattering coefficient for the terrain and geometric effects using SRTM digital elevation model at 30 m resolution, and (4) reducing the speckle effects by using Lee speckle filter. The SLC products were also used to calculate the coherence from the SNAP platform in five successive steps: (1) applying an orbit file for a better estimation of position, (2) applying back-geocoding for co-registration, (3) running the “coherence module” to estimate coherence, (4) running the “TOPSAR-Deburst” module for removing the black band in SLC products, and (5) running the “Terrain correction” module using SRTM DEM.

The incidence angle for the product is 40 degrees for both R3 and Sidi Rahal sites and 35 degrees for the Chichaoua site Amazirh et al. [21], Ouaadi et al. [31]. Note that the 20 m resolution Sentinel-1 data were finally resampled at the field-crop scale.

#### 2.2.5. Vegetation Descriptors

Vegetation descriptors depict the growth of vegetation, its density, and its impact on the radar backscatter. The accuracy and sensitivity of vegetation descriptors are essential for the correct estimation of SM throughout the agricultural season. Three different vegetation descriptors used as a proxy for vegetation in the linear and non-linear radar models are described below:

(1) NDVI is derived from Sentinel-2 surface reflectance—ESA launched the Sentinel-2 optical satellites S2-A and S2-B in 2015 and 2017, respectively. They provide optical images at 10–60 m resolution with a revisit cycle of 5 days. The Sentinel-2 level-2 product downloaded from the THEIA platform is used in this paper. Sentinel-2 product is ortho-rectified and corrected for the atmospheric effect using the MAJA processor [65] for cloud detection and atmosphere correction. NDVI is calculated from (near-infrared) band 8 and (red) band 4.

(2) PR is derived from Sentinel-1 data by taking the VH/VV polarization ratio.

(3) CO is calculated from S1A and S1B from two consecutive acquisitions of the same orbit (i.e., 6 days overpass with S1A and S1B)—the product of one SAR image with its complex conjugate of the second image from a local neighborhood [66,67].

Note that the vegetation indices NDVI, PR, and CO are resampled at the crop field scale and re-normalized by their minimum and maximum values obtained during the entire study period. The normalized NDVI, PR, and CO values thus lie in the range from 0 to 1.

### 2.3. Remote Sensing Method

#### 2.3.1. DISPATCH

DISPATCH [46,47] is one downscaling reference algorithm that combines L-band passive microwave with optical/thermal data to provide SM at the optical/thermal spatial resolution. DISPATCH was initially developed using MODIS data at 1 km resolution. The algorithm computes the soil evaporative efficiency (*SEE*) from optical/thermal data. It distributes the 1 km disaggregated resolution SM by establishing a relationship between 40 km resolution SMOS/SMAP SM and 1 km resolution *SEE* estimates. *SEE* shows a quasi linear relationship with the soil temperature [68] retrieved from optical/thermal data. Note that the DISPATCH approach assumes that the spatial variability of SM does not significantly change between the SMOS/SMAP and optical/thermal overpasses. The C4DIS processor based on the DISPATCH algorithm was developed at Centre Aval de Traitement des Données SMOS (CATDS) to provide SM at 1 km resolution on a global and daily basis.

Recent improvements in the DISPATCH algorithm (named DISPATCH${}_{veg-ext}$) allowed increasing its applicability domain by including densely vegetated areas and using the MODIS EVI instead of MODIS NDVI to increase its robustness over vegetated areas [54]. Herein, the DISPATCH${}_{veg-ext}$ algorithm is sequentially applied to 36 km resolution SMAP SM and to the 1 km resolution disaggregated SMAP SM using MODIS and Landsat LST/NDVI data, respectively. The two-step downscaling algorithm is fully described in Ojha et al. [45]. In practice, the 36 km resolution SMAP SM is disaggregated at 1 km resolution. The 1 km disaggregated data set is aggregated at an intermediate spatial resolution (ISR) of 10 km, and finally, the 10 km ISR data set is further disaggregated at 100 m resolution. The use of an ISR between the two disaggregation steps provides an optimal compromise—in terms of disaggregation efficiency at the target resolution—between scaling effects and the calibration performance of the contextual DISPATCH approach [45].

The general equation of the DISPATCH algorithm, which is implemented separately at 1 km and 100 m resolutions, is defined as follows:(1)$${SM}_{HR}={SM}_{LR}+{\left(\frac{\delta SEE}{\delta SM}\right)}_{LR}^{-1}*\;\left({SEE}_{HR}-{SEE}_{LR}\right)$$
where $SM_{LR}$ is SM at low (either SMAP or 10 km) resolution, ${SEE}_{HR}$ is *SEE* at high (either MODIS or Landsat thermal) resolution calculated from MODIS/Landsat, and ${{\left(\frac{\delta SEE}{\delta SM}\right)}_{LR}}^{-1}$ is the inverse of the partial derivative of a *SEE*(SM) model calculated at *LR*. *SEE* is considered as a linear relationship with soil temperature and is written as:(2)$$SEE_{HR}=\frac{T_{s,max}-T_{s,HR}}{T_{s,max}-T_{s,min}}$$
where $T_{s,HR}$ is the soil temperature at HR, and $T_{s,max}$ and $T_{s,max}$ the soil temperature for fully dry (*SEE* = 0) and wet conditions (*SEE* = 1). More details are given in Ojha et al. [45] and Ojha et al. [54]. Note that the 100 m resolution DISPATCH SM data are finally resampled at the field crop scale for comparison with Sentinel-1-based estimates.

#### 2.3.2. Active Microwave Radiative Transfer Models

The calibration/disaggregation scheme is tested with two formulations of radiative transfer model (RTM) for C-band radar data: an empirical multi-linear regression and a semi-empirical non-linear WCM-based formulation. The multi-linear RTM is expressed as:(3)$$\sigma_{VV,Linear}=a_{ML}*SM+b_{ML}*V+c_{ML}$$
with $a_{ML}$, $b_{ML}$ and $c_{ML}$ being three coefficients to be calibrated, and *V* the vegetation descriptor (NDVI, PR or CO). $\sigma_{VV}$ is the backscattering coefficient calculated from Sentinel-1 is expressed in decibels (dB).

The non-linear RTM is derived from the WCM model of Attema and Ulaby [37]. The radar backscatter is written as the sum of the contributions from vegetation and from the soil attenuated by vegetation. The vegetation contribution is expressed as:(4)$$\sigma_{VV,veg}=A_{VV}*V_{1}\cos\theta *\left(1-T_{VV}^{2}\right)$$
(5)$$T_{VV}^{2}=e^{-2B_{VV}V_{2}sec\theta }$$
where θ is the radar incidence angle, $T_{VV}^{2}$ is the two-way attenuation, $V_{1}$ and $V_{2}$ are the vegetation parameters and $A_{VV}$ and $B_{VV}$ are unknown coefficient parameters that are dependent on the vegetation characteristics. Here, $V_{1}$ = $V_{2}$ = *V*, where *V* can be either the normalized NDVI, normalized PR or normalized CO. In our case, the soil contribution is expressed as:(6)$$\sigma_{soil,VV}=a_{WC}*SM+c_{WC}$$
with $a_{WC}$ and $c_{WC}$ being two calibration parameters.

Using Equations (Equation 4)–(Equation 6), the non-linear RTM becomes:(7)$$\sigma_{VV,Non-linear}=A_{VV}Vcos\theta \left[\left(1-e^{-2B_{VV}Vsec\theta }\right)\right]+\left(e^{-2B_{VV}Vsec\theta }\right)\left(a_{WC}*SM+c_{WC}\right)$$
where $A_{VV}cos\theta$ and $2B_{VV}sec\theta$ are assumed to be constant over each experimental site, and are hence considered as calibration parameters and defined as $A_{VV}cos\theta$ = $b_{WC}$ and $2B_{VV}sec\theta$ = $d_{WC}$. As a result, the non-linear WCM-based RTM has four parameters ($a_{WC}$, $b_{WC}$, $c_{WC}$ and $d_{WC}$) to be calibrated:(8)$$\sigma_{VV,Non-linear}=b_{WC}V\left[\left(1-e^{-d_{WC}V}\right)\right]+\left(e^{-d_{WC}V}\right)\left(a_{WC}*SM+c_{WC}\right)$$

Note that by neglecting the vegetation impact (*V* = 0), the non-linear model has the same expression as the linear one. The $d_{WC}$ parameter represents here a non-linearity index of the vegetation impact on the radar backscatter. In the same vein, parameters a, b and c have the same meaning in both linear and non-linear RTM, so that their calibrated values over the same sites can be compared.

#### 2.3.3. Coupling DISPATCH Data with Sentinel-1-Based SM Retrieval Algorithms

Figure 2 shows the schematic diagrams for the coupling of disaggregated SM with the SM retrieval algorithms based on the Sentinel-1 backscattering coefficient explained in three different steps. In the first step, a 100 m resolution SM data set is produced by disaggregating SMAP SM data at 100 m resolution. The disaggregation procedure consists of sequentially applying DISPATCH to 36 km resolution SMAP data using MODIS data and to 1 km resolution disaggregated SM using Landsat data [45]. In the second step, the previously derived 100 m resolution SM is used as a reference data set to calibrate the C-band RTM of Equations (Equation 3) and (Equation 8). In the third step, the so-calibrated RTM are run in the inverse mode to predict SM on each Sentinel-1 overpass date. The retrieved SM values are finally evaluated using the in situ SM measurements collected concurrently with Sentinel-1 data. Note that the main advantage of such a disaggregation/calibration coupling scheme is not requiring in situ data for calibrating radar models. Different configurations of radar models are tested in terms of the linear/non-linear representation of vegetation effects on the radar signal and the nature of vegetation descriptors (either NDVI, PR, or CO).

#### 2.3.4. Calibration Parameters

The three calibration parameters ($a_{ML}$, $b_{ML}$ and $c_{ML}$) of the linear RTM of Equation (Equation 3) and the four parameters ($a_{WC}$, $b_{WC}$, $c_{WC}$ and $d_{WC}$) of the non-linear RTM of Equation (Equation 8) are calibrated using the 100 m disaggregated SMAP SM data set as a reference. To evaluate the impact of uncertainties in DISPATCH 100 m resolution data, the calibration procedure is also undertaken using in situ (instead of DISPATCH) SM. In both (calibration using DISPATCH or in situ SM) cases, and with both (linear and non-linear) RTM, three model configurations are investigated by selecting NDVI, PR or CO as the vegetation descriptor within the radar modeling.

The calibration parameters for the linear RTM are estimated from the linear regression by using the ordinary least square method. The calibration parameter for the non-linear model is calculated from the non-linear least square method that is based on the Levenberg–Marquardt algorithm [69]. The optimization algorithm used here minimizes the squared error difference between the observed and predicted backscatter coefficient. For both linear and non-linear RTM, the covariance matrix is calculated. The standard error is calculated from the covariance matrix by the square root of the diagonal. The standard error of each calibration parameter is calculated to identify the uncertainty of the retrieved calibration parameters and possible compensation effects between retrieved parameters. Note that the standard error is systematically presented in terms of percentage relative to the retrieved parameter value.

For the calibration of a, b, c and d parameters, long time series of DISPATCH and Sentinel-1 data are preferred so as to cover a wide range of surface and vegetation conditions. This is the reason why the calibration procedure is undertaken herein only over Chichaoua and Sidi Rahal sites, where data were collected for more than one year. A calibration data set is derived for each site separately, while the R3 data set is kept for the *e* of the retrieved SM using a fully independent data set. The R3 data set covers only a few months during the agricultural season and is hence not fully relevant for calibration purposes. However, the in situ SM data distributed within the 22 crop fields of the R3 perimeter will be used to assess the performance of the approach in space at different times of the season. In practice, for the R3 application, the parameters used as input to the RTM are set to the average of the parameters retrieved over Chichaoua and Sidi Rahal sites.

Regarding the calibration strategy of the non-linear RTM, attention should be paid to the compensation effects between $b_{WC}$ and $d_{WC}$ parameters (not shown). Especially, when attempting to retrieve $a_{WC}$, $b_{WC}$, $c_{WC}$ and $d_{WC}$ parameters simultaneously using the brute-force method, large relative standard errors were systematically obtained for both vegetation parameters ( $b_{WC}$ and $d_{WC}$ ), notably when the retrieved $d_{WC}$ was close to zero (low vegetation impact). This resulted in estimated percentage errors larger than 100%. Therefore, one parameter can be removed from the calibration without changing the results of the calibrated RTM model. Hence, a slightly different strategy was adopted by constraining the $b_{WC}$ parameter of the non-linear RTM to be equal to the retrieved $b_{ML}$ value in the linear RTM case. Consequently, the calibration of the RTM consists in retrieving, $a_{WC}$, $c_{WC}$ and $d_{WC}$ as free parameters while setting $b_{WC}$ equal to an a priori value as an additional constraint for solving the ill-posed problem.

## 3. Results

In this section, the calibration approach is first applied to both (linear and non-linear) RTM in different configurations: using three vegetation descriptors (NDVI, PR, and CO) with two (in situ, DISPATCH) reference SM datasets separately. Then the calibrated parameters are used to estimate SM from Sentinel-1/2 data. Finally, in situ SM collected in Sidi Rahal, Chichaoua and R3 sites are used to analyze the accuracy of predicted SM.

### 3.1. Accuracy of DISPATCH SM

The accuracy of DISPATCH 100 m resolution SM is assessed with the in situ SM data set used in this study. Although, the objective herein is not to provide a full validation of the 100 m resolution disaggregated data set, it is still useful to first assess the uncertainty of this SM product with a view to better understand its possible impact on the calibration approach.

Figure 3 shows the scatter plots between DISPATCH and in situ SM (Chichaoua, Sidi Rahal). Chichaoua and Sidi Rahal sites show a correlation coefficient (R) value of 0.81 and 0.49, respectively. The mean bias (MB) at the Sidi Rahal site is relatively large whereas it is not significant at the Chichaoua site. Such a dry bias at Chichaoua can be due to uncertainties in SMAP data, in the DISPATCH methodology and/or its ancillary data as well as possibly the spatial heterogeneity of SM at a scale finer than 100 m. To try and explain the key reasons for the differences between Sidi Rahal and Chichaoua sites, the bias between remotely sensed and in situ SM was investigated in the original SMAP, the 1 km resolution DISPATCH and the 100 m resolution DISPATCH data sets separately. The bias in the original SMAP (and 1 km DISPATCH in parenthesis) SM is −0.06 (−0.06) m${}^{3}$/m${}^{3}$ and −0.02 (−0.05) m${}^{3}$/m${}^{3}$ at Sidi Rahal and Chichaoua sites, respectively.

Having a negative bias on low-resolution SM data at the Chichoua site is fully expected as the local irrigation makes the wheat field generally wetter than the surrounding area covered by the SMAP pixel. However, observing a negative bias in low-resolution SM data at the Sidi Rahal site (and even more negative than at the Chichaoua site) is unexpected. In fact, as a dryland site surrounded by irrigated areas in the west/east and bordered by the Atlas piedmont in the south, the Sidi Rahal site should be one of the driest part within the SMAP pixel. Therefore, the strong negative bias in 100 m resolution DISPATCH SM at Sidi Rahal site is likely to be attributed to a negative bias in SMAP data over that area and/or a lack of representativeness of in situ measurements in terms of sensing depth (the surface TDR sensors are located in the 5–10 cm soil layer, whereas SMAP senses, on average, the 3–5 cm soil layer).

This brief evaluation of 100 m resolution DISPATCH SM indicates that the accuracy of the disaggregated dataset can be low due to the presence of bias. However, the disaggregated SMAP data set can capture most of the spatial variability of SM at 100 m resolution as well as the SM dynamics across the agricultural season from the early stage of the crop growth to the fully grown vegetation.

### 3.2. Evaluation of Calibration Parameters

Table 1 reports the value of calibrated parameters for Chichaoua and Sidi Rahal and their associated standard error for each (linear and non-linear) RTM, using in situ SM data sets as reference for the calibration. The calibration parameters (and their associated standard errors) of both RTM for the three vegetation descriptors (NDVI, PR, and CO) are compared.

Overall, the retrieved parameters are relatively consistent from one configuration (linear/non-linear RTM and vegetation descriptors) to the other and for both (Chichaoua and Sidi Rahal) sites. The only significant difference in the results between tested configurations is the change in the sign of the b parameter for the CO vegetation descriptor. Indeed, the backscattering coefficient is negatively correlated with NDVI and PR and positively correlated with CO. However, the absolute value is rather similar in all cases, which confirms that any of the three vegetation descriptors can be used to model the vegetation effect on a radar signal. Nonetheless, when looking at the standard error percentage on the retrieved parameters, significant differences appear between parameters and between configurations. For instance, the c parameter has a very small uncertainty (about 3%) in all cases. The uncertainty in the retrieved a parameter is relatively larger (about 13% on average for all cases), but it is quite stable and consistent in all cases. However, larger errors are obtained for the parameters b and d associated with the vegetation modeling. The overall uncertainty in b and d parameters is estimated as 39% and 47%, respectively. By removing the data associated with parameter values close to zero, the overall uncertainty drops to 21% and 18%, respectively. Interestingly, the relative error in retrieved parameters slightly varies with the vegetation descriptor used. It is minimum with PR, maximum with NDVI and intermediate with CO. This is valid for all configurations, including linear/non-linear RTM and both Chichaoua/Sidi Rahal sites. Similar to Table 1, Table 2 shows the value of the calibrated parameters but when the DISPATCH 100 m SM dataset is used as a reference in the calibration. As expected, the error in retrieved parameters increases when using remotely sensed (DISPATCH) instead of in situ SM data. For instance, the overall relative error in the c parameter is 9%, compared to 3% when using in situ data for calibration. By comparing results for the three vegetation descriptors, PR shows the best performance with an average standard error (all parameters combined) of 20% and 25% over Chichaoua and 25% and 28% over Sidi Rahal, for the linear and non-linear RTM respectively. In addition, the retrieved values using DISPATCH data are closer to the in situ case for the PR configuration compared to NDVI and CO configurations. In particular, the CO configuration seems to be very sensitive to uncertainties in the reference calibration data set, resulting in relative errors of a parameter larger than 100% (see Table 2 for the Sidi Rahal site).

Summarizing the results for two different (in situ, DISPATCH) calibration SM datasets, all three vegetation descriptors (NDVI, PR, CO) provide useful information about vegetation effects, which can be efficiently represented by a linear or a non-linear model. Nevertheless, the PR configuration shows slightly better results than the NDVI and CO configurations with smaller estimated uncertainties in the retrieved parameters, and with a better stability against errors and biases in DISPATCH data. Moreover, the consistency of parameter values indicates that DISPATCH data can be used for calibrating the RTM models, despite the errors in the remotely sensed SM that may compensate over relatively long time series. Finally, the parameter values are rather similar for both linear and non-linear RTM, so that, from the calibration point of view, both approaches are still relevant.

### 3.3. Evaluation of SM Estimates

Each calibrated parameter set is fed into the linear and non-linear RTM and the SM is derived from radar backscatter and vegetation descriptor observations by using the (linear/non-linear) RTM in the inverse mode: Formally, SM for the linear model is expressed as:(9)$$SM_{Linear}=\left(\sigma_{VV}-b_{ML}*V-c_{ML}\right)/a_{ML}$$

SM for a non-linear model is expressed as:(10)$$SM_{Non-linear}=\left(\left(\sigma_{VV}-b_{WC}*V\right)*e^{d_{WC}V}+\left(b_{WC}*V\right)-c_{WC}\right)/a_{WC}$$

The predicted SM is evaluated using in situ measurements collected over the R3 perimeter and at the Sidi Rahal and Chichaoua sites.

#### 3.3.1. Temporal Analysis

SM is evaluated using two different calibration inputs (1) in situ SM and (2) DISPATCH SM at 100 m. Figure 4 plots predicted versus in situ SM in the case where in situ SM data are used to calibrate the RTM for three different vegetation descriptors (NDVI, PR, CO). For both sites (Chichaoua and Sidi Rahal), PR shows better performance with an R in the range 0.72–0.82 and 0.74–0.80 for linear and non-linear RTM, respectively. Note that the slope (S) and the bias for the linear case are equal to 1 and zero, respectively. It is because the same in situ SM data set is used for the calibration and validation. In general, the SM results are consistent with the previous study about calibration parameters as the PR configuration consistently provides a lower uncertainty in model parameters and more accurate estimates of SM.

Similar to Figure 4, Figure 5 shows the scatter plots between predicted and in situ SM, where DISPATCH SM data at 100 m resolution are used for calibration. When using DISPATCH SM data for calibration, the NDVI configuration shows a better performance for both linear and non-linear RTM models with R in the range 0.76–0.79 and 0.77–0.80 for both sites (Chichaoua and Sidi Rahal). Statistical results are significantly poorer for the CO configuration with an R in the range 0.64–0.68 for Chichaoua and Sidi Rahal, respectively.

Note that the MB in retrieved SM is quite large for Sidi Rahal. Such an MB was not observed when using in situ SM datasets for calibration. The bias is directly attributed to the dry bias in the SMAP disaggregated SM data sets (similar dry bias observed in the accuracy assessment of DISPATCH SM datasets). In general, the statistical results using DISPATCH instead of in situ SM data are degraded, but not in all cases, since with the NDVI configuration (for linear/non-linear RTM and both sites) the correlation is improved when using DISPATCH data as a reference data set. This observation is consistent with the good performance of DISPATCH over the Chichaoua site (see Figure 3). Note that part of the errors obtained in the comparison between the remotely sensed SM product and the localized TDR measurements may be attributed to the (not measured) possible variability of soil moisture within the field. However, given that the agricultural practices (including irrigation) are homogeneous at the field scale, this effect is assumed to be minor compared to other error sources.

#### 3.3.2. Spatio-Temporal Analysis

Since the performance of the CO configuration is significantly weaker than both PR and NDVI configurations in terms of both RTM parameter uncertainties and the accuracy in predicted SM, only the NDVI and PR cases are implemented in the evaluation of the approach over the R3 study area. It is reminded that over R3, the linear and non-linear RTM are run using the parameters estimated as the average of those retrieved over Chichaoua and Sidi Rahal sites. The objective is to assess the robustness of the calibration strategy in space at different times of the season.

Figure 6 shows the scatter plots between predicted and in situ SM over R3 for spatio-temporal analysis, where in situ SM data are used to calibrate the RTM for two different vegetation descriptors (NDVI, PR). Here, the NDVI configuration shows a larger R for DOY 14, 30, and 38 in the range 0.53–0.74 by combining the quite similar result for both linear and non-linear RTM. Instead, the PR configuration shows better performance for DOY 62 and 78 with an R in the range 0.15–0.41 and 0.09–0.37 for linear and non-linear RTM, respectively. Note that the relatively poor performance on DOY 78 may be due to the time difference between SM sampling dates since ground measurements were undertaken on two successive days with one-half of the monitored crop fields on DOY 78 and the other half on the day after.

Figure 7 shows the time series of R and S between the predicted and in situ SM for both linear and non-linear RTM and for NDVI and PR vegetation descriptors separately. The graphs show that both (linear and non-linear) RTM with NDVI and PR vegetation descriptors behave similarly. R and S both increase during the early stages of the vegetation growth and then both decrease when the vegetation is fully developing. As the vegetation growth increases, the R decreases for both models, especially when the vegetation index is above 0.6 in-line with the study of Baghdadi et al. [70]. The time series of Figure 7 also illustrates that the NDVI as a vegetation descriptor shows better performance than the PR configuration during the early stage of the vegetation growth, but when vegetation is fully developed, the PR for both linear/non-linear models shows better performance. This is because NDVI is more sensitive than PR to small variabilities of vegetation cover during early stages of crops while it becomes saturated above a given threshold of biomass/LAI that is lower than the saturation threshold or PR [71,72]. The use of NDVI or PR as a vegetation descriptor is therefore a compromise between precision and accuracy in a range of crop covers.

Figure 8 shows the scatter plot between predicted and in situ SM where DISPATCH 100 m SM data are used for calibration and in situ SM data for validation. On DOY 14, 30, 38, the NDVI configuration still shows a better performance with an R in the range 0.55–0.74 and 0.55–0.70 for linear and non-linear RTM, respectively. For DOY 62 and 78, the PR configuration better performs for the linear RTM, with an R in the range 0.36–0.50, but a more inadequate performance in non-linear RTM, whereas the NDVI shows an adequate configuration for the non-linear RTM, with an R in the range 0.20–0.41. The statistical results are consistent with the results obtained by using the in situ SM datasets for calibration, but with a dry bias in predicted SM associated with the dry bias in DISPATCH data over the Sidi Rahal site.

Table 3 reports the mean (average of all dates) statistical metrics in terms of R, S, MB, RMSD, and ubRMSD between predicted and in situ SM for both (linear and non-linear) RTM with their vegetation descriptors (NDVI, PR) and for both calibration strategies (using in situ and DISPATCH SM data separately for calibration). The overall results confirm the finding that the use of NDVI or PR as a vegetation descriptor and a linear or non-linear RTM is a trade-off between precision (as described by R) and accuracy (as described by S) all along the agricultural season. Based on our data set, the linear RTM using the PR as a vegetation descriptor offers a relatively good compromise in terms of robustness all throughout the season and simplicity with only three parameters to estimate.

Note that the in situ-based calibration shows better performance than the DISPATCH-based calibration. The point is that in situ data are in general extremely sparse and often simply unavailable so that such calibration cannot be undertaken continuously in space over a range of vegetation types and soil textures.

#### 3.3.3. Gain in Accuracy at the Fine Scale Compared to SMAP

The proposed calibration-disaggregation coupling scheme applies the downscaling algorithm DISPATCH twice, uses as input a number of ancillary data sets including MODIS and Landsat data, and undertakes a calibration of a C-band radiative transfer model using Sentinel-1 backscatter data. All these successive steps are likely to introduce errors that may accumulate in cascade from the SMAP (40 km) resolution to the targeted 100 m resolution. This is all the more critical as DISPATCH introduces uncertainties through the (uncertain) downscaling relationship and through the errors in its input data. Nevertheless, downscaling algorithms especially aim to improve the spatial representativeness of SM estimates, meaning they provide fine scale information that is expected to improve the accuracy of the disaggregated data at the targeted resolution. In fact, there exists a trade-off in data disaggregation between the added uncertainty (which decreases data accuracy at the fine scale) and the improved spatial representativeness (which increases data accuracy at the fine scale). In our case, the uncertainty assessment is made more complex due to the additional calibration step, as the disaggregated data set at 100 m resolution is not the final remote sensing product. It is only used to calibrate a Sentinel-1-based SM retrieval algorithm. Therefore, the random errors in the 100 m resolution disaggregated SM data are expected to have (and this assumption is verified in this study) a relatively small impact on the final Sentinel-1-based SM product.

One key issue is hence evaluating the accuracy of the 100 m resolution SM predicted by the calibration-disaggregation coupling scheme, compared to the original 40 km resolution SMAP SM estimate. A solution is to simply compare the performance of both products at the fine scale using localized in situ measurements [55]. Figure 9 compares the overall performance at the field scale of SMAP (original data without disaggregation and without synergy with Sentinel-1), the 1 km DISPATCH SM product and the calibration-disaggregation coupling scheme over Sidi Rahal and Chichaoua sites separately. In each case, the comparison between remotely sensed SM and in situ measurements was made on a point-by-point basis by extracting the pixel overlying the site. For clarity, in all cases, the remaining bias was removed and SM values below the residual SM (estimated as 0.02 m${}^{3}$/m${}^{3}$) were set to the residual SM. It is visible that the scatter of remotely sensed SM is generally larger for the calibration-disaggregation coupling scheme, as a result of the uncertainty added by the disaggregation and calibration steps. The point is that, and it is the main rationale for developing such a multi-sensor/multi-resolution approach, the spatial representativeness of the calibrated-disaggregated SM at the field scale is clearly superior to that of the original SMAP SM. In particular, the predicted range of SM values is much closer to that of in situ measurements in the calibration-disaggregation case, as supported by a slope of the linear regression between satellite and in situ SM significantly closer to 1. Note that the slope of the linear regression is a very relevant metric for evaluating the spatial representativeness of remote sensing data at a range of spatial scales because it combines both the correlation and the spatial variability indicators [55]. In all cases (both sites and linear/non-linear models), the calibrated-disaggregated outperforms the original SMAP SM estimates in terms of correlation coefficient, slope of the linear regression and unbiased root mean square error between satellite and in situ SM. Consequently, the added uncertainty due to the disaggregation and calibration steps is relatively small and still acceptable at 100 m resolution compared to the huge SM spatial variability over our study area, which is not represented by SMAP data.

## 4. Conclusions

An original approach is developed to provide SM products at a high spatio-temporal resolution based on a synergy between multi-resolution passive microwave, active microwave, and optical/thermal data. It is a three-step procedure in which (1) the 36 km resolution SMAP SM is disaggregated by DISPATCH at 100 m resolution using MODIS/Landsat optical/thermal data on clear sky days, (2) the 100 m resolution disaggregated SM data set is used to calibrate a radar-based SM retrieval model and (3) the so-calibrated radar model is run at field scale on each Sentinel-1 overpass. The calibration-disaggregation coupling approach is tested using two radar models (an empirical linear regression model and a non-linear semi-empirical formulation derived from the water cloud model) and three different vegetation descriptors (NDVI, PR, and CO). The retrieved SM values are finally evaluated over three sites (Chichaoua, Sidi Rahal and R3 perimeter) in central Morocco using the in situ SM measurements collected concurrently with Sentinel-1 data. A calibration data set is derived for Chichaoua and Sidi Rahal sites separately, where data were collected for more than one year, while the R3 data set is kept for the evaluation of the retrieved SM using a fully independent data set.

An evaluation of the retrieved parameters of the linear and non-linear RTM indicates that all three vegetation descriptors (NDVI, PR, CO) provide useful information about the vegetation effects on the radar signal, which can be efficiently represented in terms of model calibration, by a linear or a non-linear RTM. However, the PR configuration shows slightly better results than the NDVI and CO configurations with smaller estimated uncertainties in the retrieved parameters, and with a better stability against errors and biases in DISPATCH SM data.

In terms of predicted SM estimates, the temporal analysis (over Chichaoua and Sidi Rahal sites) indicates that the PR configuration shows better performance when in situ SM datasets are used for calibration. In contrast, the NDVI configuration shows slightly better performance when DISPATCH SM datasets are used for calibration. The spatio-temporal analysis (over R3 irrigated site) indicates that the NDVI configuration performs better (when calibration is undertaken using in situ or DISPATCH SM datasets), especially during the early stages of the agricultural season. However, the PR configuration still provides a more robust vegetation descriptor all throughout the agricultural season, consistent with the lower uncertainty in modeled parameters. In terms of linearity of the RTM, results are found very similar for both linear and non-linear formulations, so that the added complexity of the non-linear representation is not clearly justified by our data set.

The new calibration/disaggregation coupling scheme presented here does not require in situ data and has thus a strong potential for providing good quality and frequent SM data at a field scale over large areas for agricultural purposes. Despite the satisfying results obtained over the study sites in this paper, the 100 m resolution pixel may be relatively low compared to the typical size of crop fields in many regions. Two avenues are foreseen to help increase this resolution. The first avenue would be to consider the higher spatial resolution of Sentinel-1 data: the RTM parameters obtained at 100 m resolution from the synergy between DISPATCH and Sentinel-1 data, as presented in this paper, could be used at 20 m resolution to invert SM from Sentinel-1 at 20 m resolution. The second avenue would be to prepare for the near future advent of thermal infrared sensors with improved capacities in terms of spatial and temporal resolutions such as the TRISHNA mission [73]. In addition, a characterization of the error sources coming from the original SMAP data and from the DISPATCH methodology is necessary to assess their impact in the calibration procedure over a wider range of conditions. It is further reminded that both DISPATCH and Sentinel-1 data have intrinsic limitations, including the impact of cloud coverage (DISPATCH) and the lack of SM sensitivity over densely vegetated pixels (both DISPATCH and Sentinel-1). Nonetheless, further validation is needed to cover various incidence angles of Sentinel-1, crop types, and surface conditions in the future. In addition, a characterization of the error sources coming from the original SMAP data and from the DISPATCH methodology is necessary to assess their impact in the calibration procedure over a wider range of conditions. Furthermore, attention is drawn to one assumption underlying the temporal calibration strategy of the radar model: parameters were assumed to be constant in our study. The point is that soil roughness and vegetation parameters are likely to vary in time as a result of agricultural practices (notably, plowing and crop rotation). Therefore, future studies should investigate the dynamical retrieval of model parameters over periods long enough to allow for an efficient calibration using a sufficient number of DISPATCH SM estimates concurrently with Sentinel-1 overpasses and, at the same time, short enough to represent possible changes in model parameters.

## Figures and Tables

**Figure 1 sensors-21-07406-f001:**
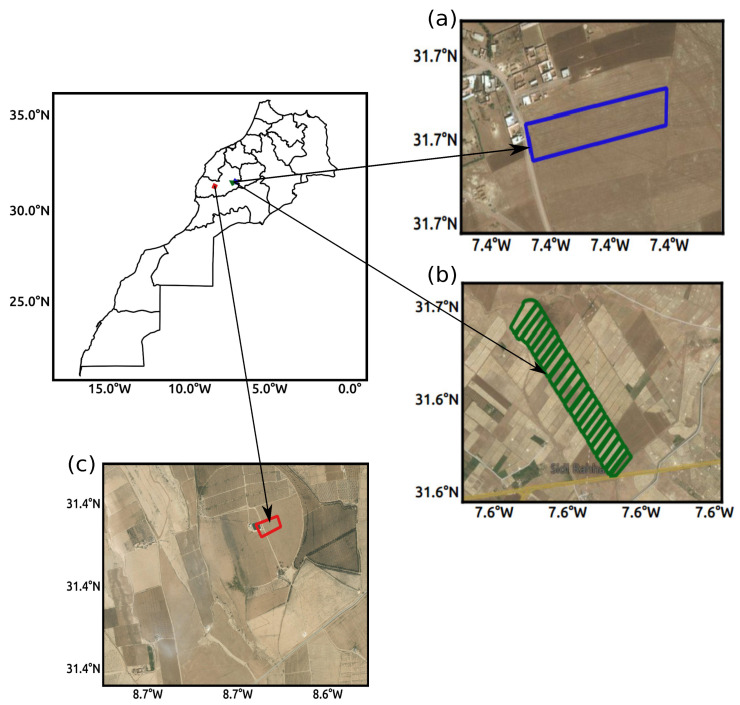
Location of the study area, including (**a**) Sidi Rahal, (**b**) R3 irrigated perimeter, (**c**) Chichaoua experimental sites.

**Figure 2 sensors-21-07406-f002:**
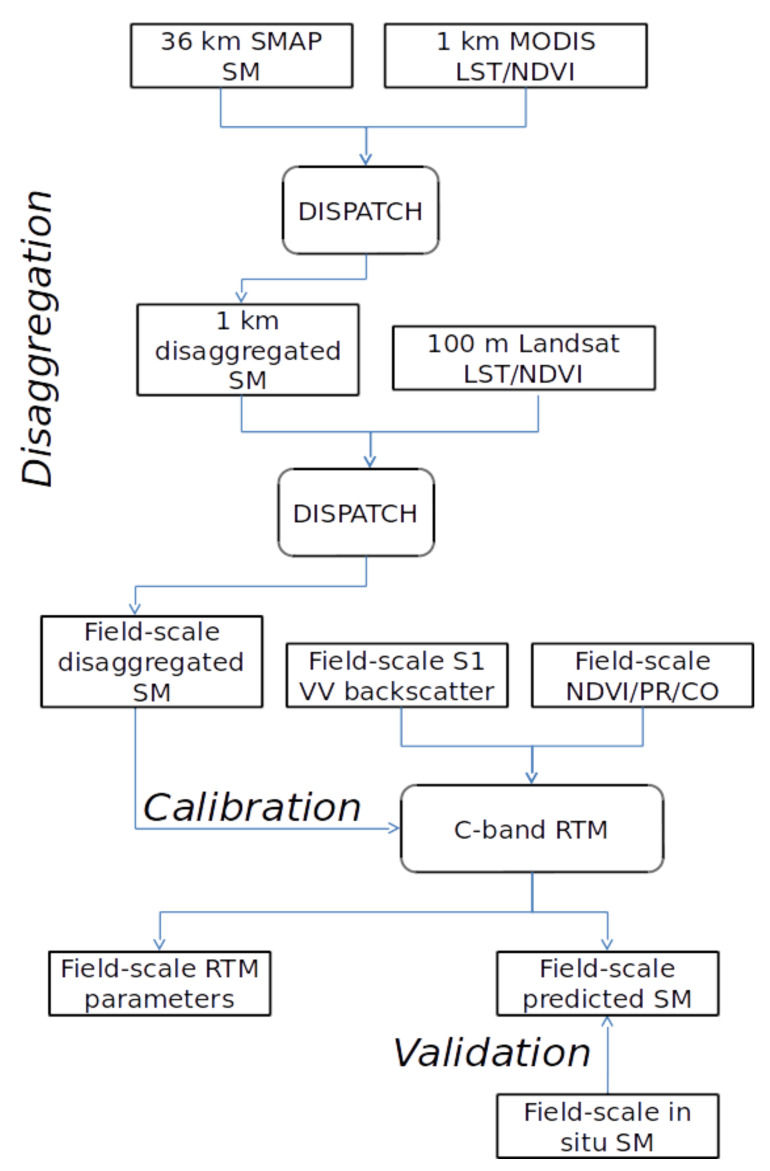
Schematic diagram of the calibration/disaggregation coupling scheme combining SMAP passive microwave, MODIS/Landsat optical/thermal and Sentinel-1 radar data at a range of spatial resolutions to provide an SM product at the field scale at the temporal frequency of Sentinel-1.

**Figure 3 sensors-21-07406-f003:**
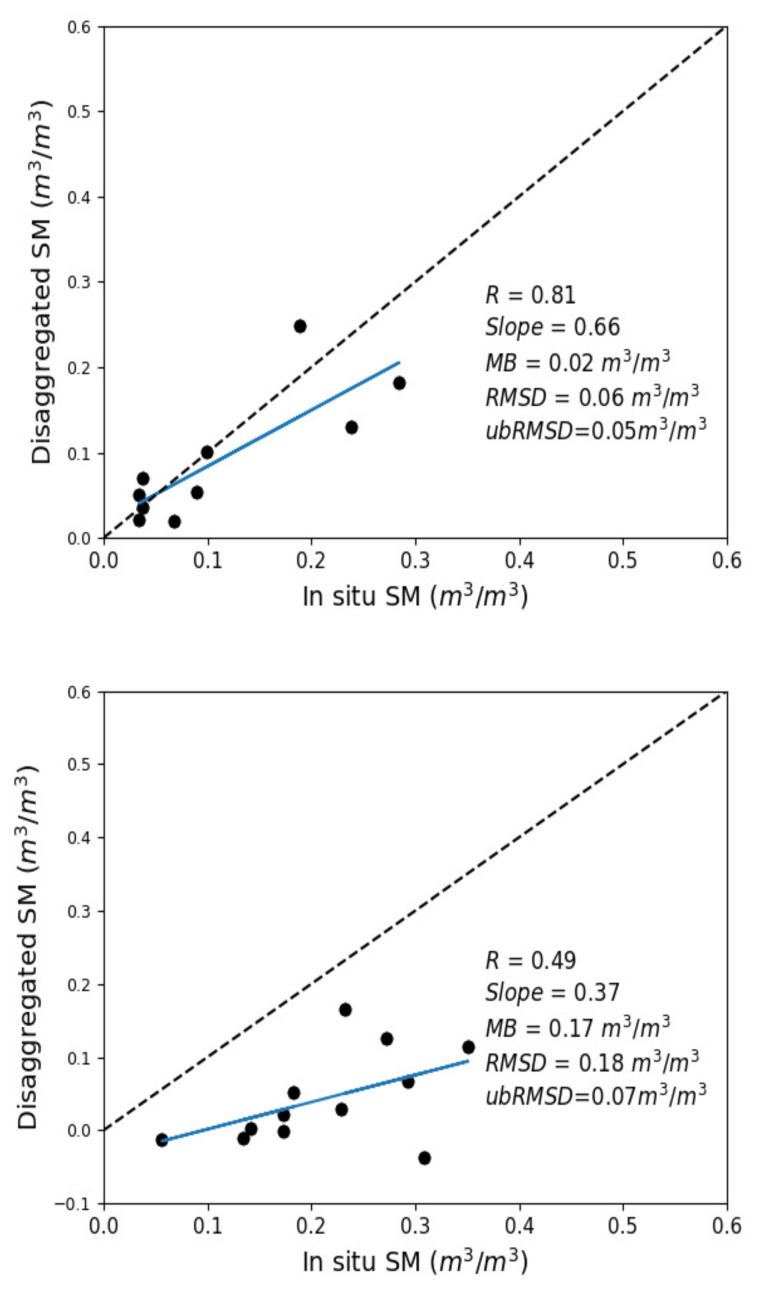
DISPATCH 100 m resolution disaggregated SM versus in situ measurements at Chichaoua (**top**) and Sidi Rahal (**bottom**) sites.

**Figure 4 sensors-21-07406-f004:**
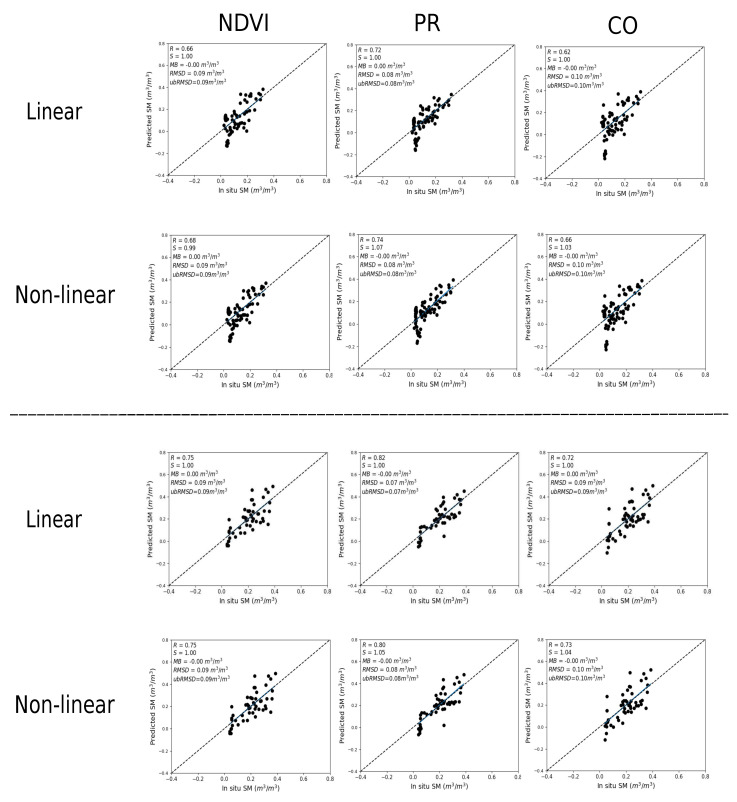
Scatter plot over Chichaoua (**top**) and Sidi Rahal (**bottom**) of predicted versus in situ SM, for linear and non-linear RTM with NDVI, PR and CO as vegetation descriptors separately where in situ SM data are used for calibration.

**Figure 5 sensors-21-07406-f005:**
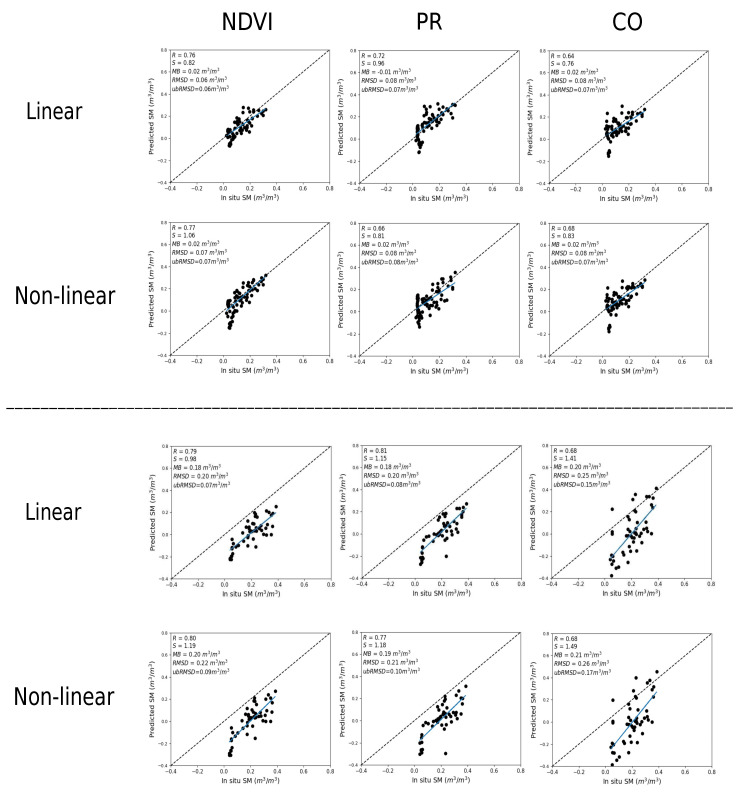
Same as Figure 4 but with DISPATCH SM data used for calibration.

**Figure 6 sensors-21-07406-f006:**
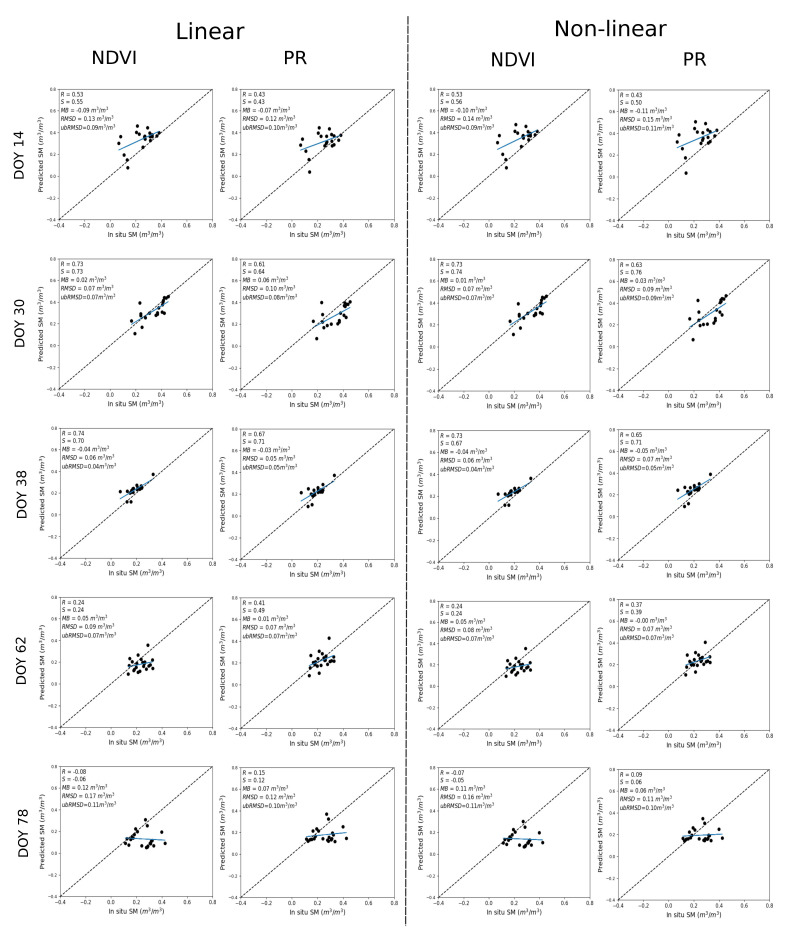
Scatter plot over the R3 irrigated parameter of predicted versus in situ SM, for linear (**left**) and non-linear (**right**) RTM with NDVI and PR as a vegetation descriptor separately where in situ SM data are used for calibration.

**Figure 7 sensors-21-07406-f007:**
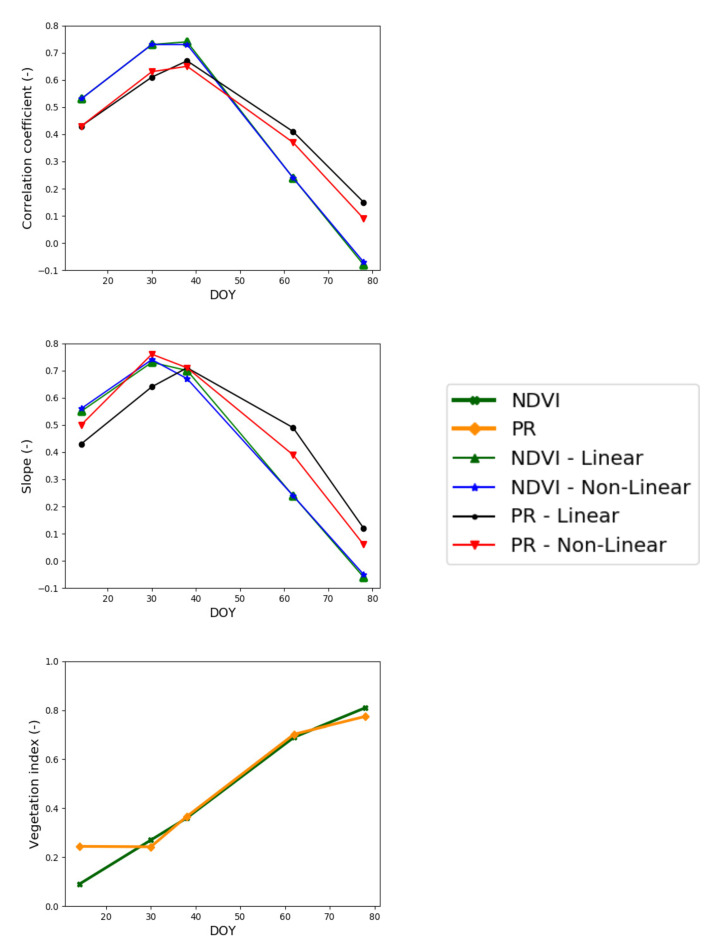
Time series of the correlation coefficient and slope of the linear regression between predicted and in situ SM for both (linear and non-linear) RTM, two vegetation descriptors (NDVI and PR) separately, and where in situ SM datasets are used for the calibration.

**Figure 8 sensors-21-07406-f008:**
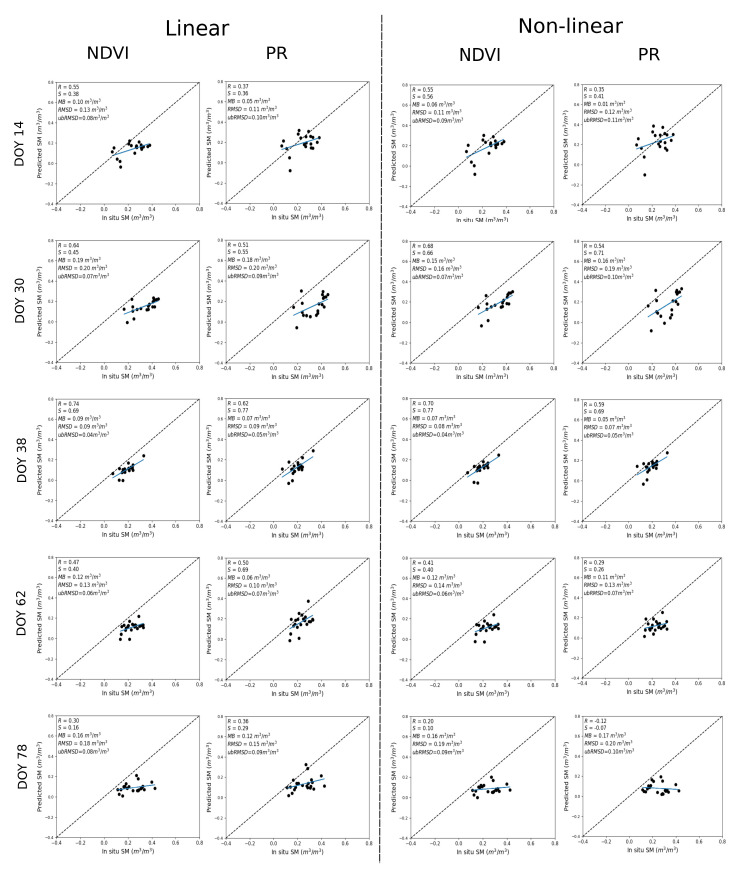
Same as Figure 6 but with DISPATCH SM data used for calibration.

**Figure 9 sensors-21-07406-f009:**
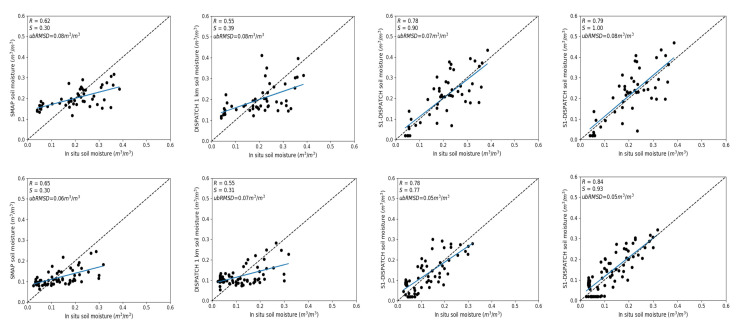
Scatter plots (from left to right) of the original 40 km resolution SMAP SM, the 1 km resolution DISPATCH SM, the 100 m resolution calibrated-disaggregated SM for the linear model and the 100 m resolution calibrated-disaggregated SM for the non-linear model versus in situ measurements, for Sidi Rahal (**top**) and Chichaoua (**bottom**) sites separately.

**Table 1 sensors-21-07406-t001:** Values of the calibration parameters and their standard error for linear and non-linear RTM using NDVI, PR and CO as vegetation descriptor separately for Chichaoua (left) and Sidi Rahal (right) sites where in situ SM data are used for the calibration.

		Vegetation Descriptors	Chichaoua	Sidi Rahal
Linear	Calibration parameter $a_{ML}$(dB*$m^{3}/m^{3}$)/$b_{\mathrm{ML}}\left(\mathrm{dB}\right)/c_{\mathrm{ML}}$(dB)	NDVI	19/−3/−14	20/−0.7/−18
		PR	16/−6/−12	20/−4/−16
		CO	15/3/−16	17/3/−19
	Standard error percentage std${}_{a_{ML}}$(%)/std${}_{b_{ML}}$(%)/std${}_{c_{ML}}$(%)	NDVI	17/28/2	15/124/3
		PR	12/11/3	10/21/3
		CO	16/18/3	14/30/3
Non-linear	Calibration parameter $a_{WC}$(dB*$m^{3}/m^{3}$)/$b_{WC}$(dB)/$c_{WC}$(dB)/$d_{WC}$(dB)	NDVI	18/−3/−14/−0.28	19/−0.7/−18/−0.04
		PR	11/−6/−11/−0.9	17/−4/−16/−0.4
		CO	18/3/−17/0.2	18/3/−19/0.15
	Standard error percentage std${}_{a_{WC}}$(%)/std${}_{b_{WC}}$(%)/std${}_{c_{WC}}$(%)/std${}_{d_{WC}}$(%)	NDVI	13/28/2/21	14/124/3/175
		PR	11/11/4/12	11/21/4/26
		CO	14/18/3/14	14/30/4/33

**Table 2 sensors-21-07406-t002:** Same as Table 1 where DISPATCH SM data are used for the calibration.

		Vegetation Descriptors	Chichaoua	Sidi Rahal
Linear	Calibration parameter $a_{ML}$(dB*$m^{3}/m^{3})/b_{ML}$(dB)/$c_{ML}$(dB)	NDVI	34/−7/−14	26/−4/−12
		PR	19/−9/−11	19/−6/−12
		CO	23/5/−18	11/4/−16
	Standard error percentage std${}_{a_{ML}}$(%)/std${}_{b_{ML}}$(%)/std${}_{c_{ML}}$(%)	NDVI	27/46/6	38/47/10
		PR	29/24/8	41/27/8
		CO	28/29/7	121/67/9
Non-linear	Calibration parameter $a_{WC}$(dB*$m^{3}/m^{3}$)/$b_{WC}$(dB)/$c_{WC}$(dB)/$d_{WC}$(dB)	NDVI	20/−7/−13/−0.76	19/−4/−12/−4
		PR	10/−9/−10/−2	14/−6/−11/−0.78
		CO	25/5/−18/0.27	12/4/−16/0.24
	Standard error percentage std${}_{a_{WC}}$(%)/std${}_{b_{WC}}$(%)/std${}_{c_{WC}}$(%)/std${}_{d_{WC}}$(%)	NDVI	25/46/6/41	41/47/11/54
		PR	32/24/12/31	44/27/9/31
		CO	29/29/8/30	122/67/9/67

**Table 3 sensors-21-07406-t003:** Mean statistical results over 22 parcels of R3 irrigated sites in terms of correlation coefficient (R), slope of the linear regression (S), mean bias (MB), RMSD, and ubRMSD between predicted and in situ SM, for linear and non-linear RTM with NDVI and PR as a vegetation descriptor and where in situ (left) and DISPATCH (right) SM datasets are used for calibration separately.

Calibration		In Situ SM Datasets					DSIAPTCH SM Datasets				
Model		R (-)	Slope (-)	MB (m${}^{3}$/m${}^{3}$)	RMSD (m${}^{3}$/m${}^{3}$)	ubRMSD (m${}^{3}$/m${}^{3}$)	R (-)	Slope (-)	Absolute MB (m${}^{3}$/m${}^{3}$)	RMSD (m${}^{3}$/m${}^{3}$)	ubRMSD (m${}^{3}$/m${}^{3}$)
Linear	NDVI	0.43	0.43	0.01	0.10	0.08	0.54	0.42	0.13	0.15	0.07
	PR	0.45	0.48	0.01	0.10	0.08	0.47	0.53	0.10	0.13	0.08
Non-linear	NDVI	0.43	0.43	0.01	0.10	0.08	0.51	0.50	0.11	0.14	0.07
	PR	0.43	0.48	0.01	0.10	0.08	0.33	0.40	0.10	0.14	0.09

## Data Availability

Not applicable.

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
