# Peer review of "A Calibration/Disaggregation Coupling Scheme for Retrieving Soil Moisture at High Spatio-Temporal Resolution: Synergy between SMAP Passive Microwave, MODIS/Landsat Optical/Thermal and Sentinel-1 Radar Data"

_sensors, 2021, doi:10.3390/s21217406_

Round 1

Reviewer 1 Report

Soil moisture detection by using remote sensing tools is quite difficult and limited due to the reduced sensing depth (3-5 cm) to determine this parameter, among other physical item as soil roughness. In this sense, the proposed article is a good contribution to understand this and moreover, it is of high interest to compare several data with different spatial resolution like SMAP, MODIS, Landsat and Sentinel, but this is not a new strategy.

Optimization of irrigation based on the control of soil moisture should be indicated as a target of this article, in my opinion is the leiv motiv of the implementation of the algorithm presented by the authors. The use of water for agricultural pourposes and the adequate management should be the basement of the determination of soil moisture in agricultural fields. In this sense, some references may be given in the introduction regarding this.

The disaggregation from close to 40 km to only 20 m, is a laborious and complex process, the accuracy may be not so satisfactory. The use of DISPATCH is questioned, and I am not sure that the results are enough good to be published. Please see the articleDOI:10.5194/hess-2018-94 (https://hess.copernicus.org/preprints/hess-2018-94/hess-2018-94.pdf)

, which has a similar strategy and similar experimental methodology that those exposed in the article. Moreover, recently, an article has been published following a similar methodology for the South of France using SMAP, MODIS and Sentinel-3 data (https://doi.org/10.3389/fenvs.2021.555216).

Several articles applied the DISPATCH in Morocco with limited satisfactory results (i.e. https://doi.org/10.3390/rs70403783 and https://doi.org/10.3390/rs9111155)

Formally, the article is well written and structured. However, I think that not new information is given comparing with other works published. It would be interesting to remark what is new and different from previous works. Please, in my opinion, it is important to remark and show the differences from previous works to publish this article. I apologise if authors disagree with this.

Reviewer 2 Report

This study presents a solid analysis in the field of spatial downscaling for surface soil moisture and the result shows good potential for future application at high-resolution scales (below 100m). The manuscript is well rewritten and clearly expressed. A minor revision is required before acceptance.

The references of the disaggregation approaches are not enough. Recent research progresses present good direction for soil moisture downscaling and they should be added, such as:

https://doi.org/10.1016/j.jhydrol.2021.126930

https://doi.org/10.1016/j.jhydrol.2020.125894

https://doi.org/10.1016/j.rse.2021.112301

The data introduction sections and the method introductions section should be grouped into separated groups to give a clear introduction.

The explanation about the dry bias in Sidi Rahal site is not enough. What are the key reasons for the differences between these two sites?

The explanation about the poor performance in Figure 6 and 8 should be enhanced.

In Figure 9, how to conduct the comparison between in situ measurements and the downscaled values? The expression in current version is not clear.

The authors have introduced two models for SM estimation with the in cooperation of Sentinel SAR data: Linear and non-linear model. In your abstract and conclusion part, the result should be indicated for the readers to the right selection. Meanwhile, the non-linear model should outperform linear model in some times. How to explain the difference?

Round 2

Reviewer 1 Report

First of all, thanks for the response point by point to my comments and thanks to disagree with some comments. Believe me, it is important for reviewers to have this type of comments too. This help me to a better understanding of the work done. I think that the article has been improved enough to be published. 

I believe that it is important to indicate the implementation of DISPATCH at 100 m resolution. Obviously, we can discuss about it and the relative importance for water management, depending on the size of the area affected by irirgation and other agronomical considerations, but it is true that this is a good step for retrieving soil moisture at a higher resolution than it is commonly studied. In this sense, congratulations.

This manuscript is a resubmission of an earlier submission. The following is a list of the peer review reports and author responses from that submission.

Round 1

Reviewer 1 Report

please refer to the attached file for my comments 

Reviewer 2 Report

Please, see attached pdf file.

Reviewer 3 Report

The objective of the article presented to be reviewed is not clear defined and in general, it is difficult to read and understand what the authors want to show. As a suggestion, the objetctive(s) should be clearly defined.

The introduction is mainly centred in the description of the missions and data used, but it has no information about soil and the properties that are key in the soil moisture (SM). Moreover, the water management, considering agricultural areas, is very important for SM.

In this sense, some soil parameters like organic matter content or colour, would be of interest. In fact, the methods used for soil sampling, soil texture, location of the samples, …, and references of the methods of analysis used are desirable. Only two TDR sensors to check the moisture of a field of such size may be not enough. It is very important a high density of sampling points in order to obtain good results. Moreover, when the work is about moisture.

The process to obtain a final pixel size of 100 m would be adequate, although the origin is a 36 km. I wonder about the mistakes that would happen during resizing, the dissagregation process. However, the significant for agricultural management of a 100 m pixel may be low. We are talking a 1 ha pixel size which will be so large in most of the Mediterranean cultivations, for instance.

I have some doubts about the strategy followed when authors talk about making equal bwc parameter with bml. I apologise but the reasons given do not seem convincing to me.

I think that article need some discussion and be more critical with the results. In part, between lines376 to 380, authors indicate something about the limitations of the method.

The comparison between soil data (real SM) and data calculated applied this methodology, in my opinion, is not accurate. Poor data from soils, low density of points and large pixel size to compare.

I understand that this article opens an interesting line but with these results, I suggest that we cannot conclude the possibility of measurement SM in situ without calibration.

Please check the separation of the word at the end of line 108, 225.

Please check the lines 208 and 209 “bamler1998synthetic…”. Probably authors want to add the references of Bamler and Hartl (1998) and Touzi (1999).

Reviewer 4 Report

This manuscript presents an interesting study with the aim for calibrating sentinel-1 radar to regional soil moisture. However, about the calibration strategy, a key issue should be answered in advance. The authors have tested the results from the model calibrated with in-situ soil moisture, why introduce disaggregated soil moisture into the calibration? Usually, the in-situ observations should be with good accuracy than disaggregated SM from passive microwave observations. To this sense, your calibration with the use of the disaggregated SM is not meaningful. In your manuscript, you also have conducted a comparison study to investigate the performances from in-situ based calibration model and disaggregated based calibration model. The in-situ based shows better performance. Meanwhile, your calibration was only conducted with the observations from a single plot, how to ensure the representativeness of this observation to different landscapes. Land cover difference will introduce variations in vegetation type and soil texture.

About your purpose, according to your assumption, the disaggregated SM should be with good accuracy. Therefore, why not directly use the downscaling SM under clear-sky condition to get the fine-scale SM but redirect to sentinel estimation?

During your downscaling, why not directly use the 1-km downscaled results but swithc to the intermediate spatial resolution? How to consider the impact from the scale difference? Have you tested the results from the direct downscaling and the us e of the ISR.

In your calibrated model, you only provide the error associated with each parameter but not the general performance of the model. The result is also very important for the readers.